# Antimicrobial Activity of Water-Soluble Silver Complexes Bearing C-Scorpionate Ligands

**DOI:** 10.3390/antibiotics13070647

**Published:** 2024-07-13

**Authors:** Abdallah G. Mahmoud, Sílvia A. Sousa, M. Fátima C. Guedes da Silva, Luísa M. D. R. S. Martins, Jorge H. Leitão

**Affiliations:** 1Centro de Química Estrutural, Institute of Molecular Sciences, Instituto Superior Técnico, Universidade de Lisboa, Av. Rovisco Pais, 1049-001 Lisbon, Portugal; abdallah.mahmoud@tecnico.ulisboa.pt; 2Department of Chemistry, Faculty of Science, Helwan University, Ain Helwan, Cairo 11795, Egypt; 3Department of Bioengineering (DBE), Institute for Bioengineering and Biosciences (iBB), The Associate Laboratory Institute for Health and Bioeconomy (i4HB), Instituto Superior Técnico (IST), Universidade de Lisboa, 1049-001 Lisboa, Portugal; sousasilvia@tecnico.ulisboa.pt; 4Departamento de Engenharia Química, Instituto Superior Técnico, Universidade de Lisboa, Av. Rovisco Pais, 1049-001 Lisboa, Portugal

**Keywords:** silver, complex, water-soluble, C-scorpionate, antimicrobial, antifungal

## Abstract

The novel hydrosoluble silver coordination polymer [Ag(NO_3_)(μ-1κ*N*;2*κN*′,*N*″-TPM^OH^)]_n_ (**1**) (TPM^OH^ = tris(1*H*-pyrazol-1-yl)ethanol) was obtained and characterized. While single crystal X-ray diffraction analysis of compound **1** disclosed an infinite 1D helical chain structure in the solid state, NMR analysis in polar solvents confirmed the mononuclear nature of compound **1** in solution. Compound **1** and the analogue [Ag(μ-1κ*N*;2*κN*′,*N*″-TPMS)]_n_ (**2**) (TPMS = tris(1*H*-pyrazol-1-yl)methane sulfonate) were evaluated with regard to their antimicrobial activities towards the Gram-negative *Escherichia coli*, *Pseudomonas aeruginosa*, and *Burkholderia contaminans*, the Gram-positive *Staphylococcus aureus*, and the fungal species *Candida albicans* and *Candida glabrata*. Compound **1** exhibited minimal inhibitory concentration (MIC) values ranging from 2 to 7.7 µg/mL towards the tested Gram-negative bacteria, 18 µg/mL towards the Gram-positive *S. aureus*, and 15 and 31 µg/mL towards *C. albicans* and *C. glabrata*, respectively. Compound **2** was less effective towards the tested bacteria, with MIC values ranging from 15 to 19.6 µg/mL towards the Gram-negative bacteria and 51 µg/mL towards *S. aureus*; however, it was more effective against *C. albicans* and *C. glabrata*, with MIC values of about 6 µg/mL towards these fungal species. The toxicity of compounds **1** and **2** was assessed by evaluating the survival of the *Caenorhabditis elegans* model organism to concentrations of up to 100 µg/mL. The value of 50% lethality (LD_50_) could only be estimated as 73.2 µg/mL for compound **1** at 72 h, otherwise LD_50_ was >100 µg/mL for both compounds **1** and **2**. These results indicate compounds **1** and **2** as novel silver complexes with interesting antimicrobial properties towards bacterial and fungal pathogens.

## 1. Introduction

The global rise in antimicrobial-resistant human pathogens has become a critical concern worldwide. This emergence of antimicrobial resistance stems from several factors, including the misuse and abuse of antimicrobials, allied with a decreased investment by pharma on the development of novel antimicrobials [1]. In 2017, the World Health Organization (WHO) issued the first list of antimicrobial-resistant bacterial pathogens, prioritized as of critical, high, and medium urgency [2]. The WHO list was prepared for the guidance and promotion of research and development of new antimicrobials to surpass the worldwide crisis of antimicrobial resistance. The shortage of new antimicrobials coming into market, together with the increasing resistance, led to the search for novel antimicrobials, an effort mainly carried out by the academia [2]. This effort led to the discovery of novel molecules with antimicrobial activity, including peptides [3], extracts from plants, herbs and spices [4], polymeric materials modified with molecules known for their antimicrobial activities [5], organic molecules, and metal-based complexes [6].

Silver complexes have emerged as promising antibacterial agents in various therapeutic protocols due to their potent antimicrobial properties coupled with low human toxicity [7,8,9]. They possess broad-spectrum antimicrobial activity, effectively targeting both Gram-positive and Gram-negative strains, along with resistance mitigation capabilities and the ability to disrupt bacterial biofilms [10,11]. With their favorable safety profile, silver complexes facilitate wound healing and exhibit synergistic interactions with other antimicrobial agents to optimize therapeutic outcomes [12,13]. Although the precise mechanism of action of silver-based pharmaceuticals remains incompletely understood, their efficacy involves the release of Ag(I) ions, which can penetrate cell membranes and disrupt their functionality [14]. This disruption is thought to result from non-specific mechanisms like DNA and protein binding, generation of reactive oxygen species, interfering with membrane functions and energy generation processes, as well as with efflux pumps [15,16]. Consequently, it is essential to utilize ligands that can efficiently coordinate with the active Ag(I) ions to enhance their therapeutic potential [17]. In this context, several classes of ligands have been studied, including N-heterocyclic carbenes [18,19,20], carboxylates [21,22,23], phosphines [24,25,26], and N-ligands [27,28,29,30]. Meeting the growing need for effective therapeutic agents, the development of water-soluble Ag complexes possessing antiproliferative or antibacterial activity is increasing. 

Tris(1*H*-pyrazol-1-yl)methane (TPM, Figure 1) and its derivatives, known as C-homoscorpionates, or just C-scorpionates, represent an important type of N-donor tridentate class of ligands that have garnered significant attention in the field of coordination chemistry [31,32,33,34]. The water-soluble TPM derivatives, tris(1*H*-pyrazol-1-yl)methane sulfonate (TPMS) and tris(1*H*-pyrazol-1-yl)ethanol (TPM^OH^), have been utilized to obtain hydrophilic metal complexes for biological applications or catalytic transformations in an aqueous medium (Figure 1 top) [35]. The high versatility of these compounds is attributed to the possibility of interchanging between the *N*,*N*-bipodal and the *N*,*N*,*N*- or *N*,*N*,*O*-tripodal modes of coordination (Figure 1 bottom). Additionally, they address the limitations of poly(pyrazol-1-yl)borates, which are unstable towards hydrolysis and, despite their ionic nature, insoluble in water [35].

While the coordination chemistry of hydrosoluble C-scorpionates with group 11 transition metals has been extensively focused on copper [35,36], that of silver is limited to [Ag(μ-1*κN*;2*κN*′,*N*″-TPMS)]_n_, [Ag(TPMS)(P)](BF_4_), and [AgP_4_]·4TPMS·BF_4_ examples (P = phosphine ligand) [30,37,38]. Silver complexes based on TPM^OH^ have not been reported yet; therefore, filling this gap constitutes one of the objectives of the current study.

Despite their high-water solubility and stability against hydrolysis, investigations on the antiproliferative and antibacterial activities of metal complexes bearing hydrophilic C-scorpionates have been restricted to a few studies. Silver(I)-TPMS complexes have displayed significant antibacterial and antifungal activities against a set of microbial pathogens, including *Candida albicans*, *Pseudomonas aeruginosa*, *Staphylococcus aureus*, *Enterococcus faecalis*, *Escherichia coli*, *Streptococcus pneumoniae*, *Streptococcus pyogenes*, *Streptococcus mutans*, and *Streptococcus sanguinis* [29,30]. The antiproliferative activity of [Ag(TPMS)] has been studied against two human tumor cell lines, A2780 and HCT116 [37]. The compound exhibited a significantly stronger effect on A2780 cells, with an IC_50_ of 0.04 μM, compared to HCT116 cells and normal fibroblasts. Concerning the other metal complexes based on TPM^OH^, only the water-soluble cobalt(II) complex, [Co(TPM^OH^)_2_]·[Co(TPM^OH^)(H_2_O)_3_]_2_(Cl)_6_, has been tested, exhibiting moderate in vitro cytotoxic activity against the human HepG2 hepatocellular carcinoma and HCT116 colorectal cell lines [39].

Herein, we report the synthesis and characterization of the novel silver coordination polymer bearing the TPM^OH^ ligand, [Ag(NO_3_)(μ-1*κN*;2*κN*′,*N*″-TPM^OH^)]_n_ (**1**). The antimicrobial activity of the silver–TPMS and –TPM^OH^ complexes was investigated against representative strains of the Gram-negative bacteria, *E. coli*, *P. aeruginosa*, and *Burkholderia contaminans*, the Gram-positive *S. aureus*, and the fungal species, *C. albicans* and *Candida glabrata*. The toxicity of both compounds was also assessed using living *Caenorhabditis elegans* by means of evaluating L4 larvae survival in the presence of concentrations of up to 100 µg/mL of each compound. 

## 2. Results and Discussion

Reacting equimolar amounts of TPM^OH^ and AgNO_3_ in methanol resulted in the formation of complex [Ag(NO_3_)(μ-1κ*N*;2*κN*′,*N*″-TPM^OH^)]_n_ (**1**) as colorless crystals, which were stable in air and soluble in water, alcohols, and dimethylsulfoxide (DMSO). Complex [Ag(TPMS)]_n_ (**2**) was obtained by reacting the lithium salt of TPMS with AgNO_3_ in methanol at room temperature, following our published procedure [37]. The solid-state molecular structure of compound **1** was established using single crystal X-ray diffraction (SCXRD) analysis.

Compound **1** crystallized in the monoclinic space group *Cc*, with the asymmetric unit comprising one TPM^OH^, a silver cation, a nitrate anion, and a crystallization water molecule. The infinite 1D helical chain of the compound was revealed upon symmetry expansion, disclosing the TPM^OH^ ligand bridging two silver cations by acting as an NN-donor to one metal and as an N-donor to another (Figure 2 and Figure 3) thus forming the (type 1) coordination polymer (base vector [101]; Appendix A). Each Ag metal center in **1** adopted a distorted *N*_3_*O* tetrahedral geometry, with Ag–N bond lengths falling within the range of 2.231(6) to 2.452(4) Å, an Ag–O bond length of 2.57(4) Å, and an intrachain Ag···Ag distance of 5.0830(6) Å. Hydrogen bond interactions with graph set $D_{1}^{1}\left(2\right)$ (Appendix A) involved the TPM^OH^ hydroxyl group as a donor towards the crystallization water molecule which, in turn, donated to the nitrate ligand. The consequent formation of the $C_{2}^{2}$(12) and $C_{2}^{2}$(6) graph sets explained the expansion of the structure to a (type 3) 3D framework (base vectors: [100], [001], [010]).

The high solubility of compound **1** in polar solvents, such as water and DMSO, suggests the potential dissociation of its polymeric structure upon dissolution. This hypothesis can be substantiated through NMR measurements in DMSO-*d*_6_ (Appendix A). Both ^1^H and ^13^C NMR spectra of compound **1** exhibit a set of resonances like those of TPM^OH^ pro-ligand. The ^1^H NMR spectrum of compound **1** reveals that all pyrazolyl protons are equivalent and downfield shifted when compared to those of free TPM^OH^ (Figure 4) due its coordination to silver. Similarly, the ^13^C NMR spectrum shows a slight downfield shift of all pyrazolyl carbons upon coordination (Appendix A). Therefore, the NMR analysis suggests that compound **1** in solution exhibits a monomeric complex structure where the scorpionate ligand adopts an *NNN* coordination mode (Figure 5). The same behavior is also observed and reported for **2**, which exhibits a polymeric structure of [Ag(μ-1κ*N*;2*κN*′,*N*″-TPMS)]_n_ in the solid state and dissociates in polar solvents [37].

Compounds **1** and **2** showed a different efficacy towards the bacterial and fungal strains tested (Table 1), with compound **1** showing the highest antimicrobial activity towards bacteria than **2**. On the other hand, compound **2** was more active towards the two *Candida* strains tested (Table 1). The MIC of compound **1** towards *P. aeruginosa* (2.0 μg/mL = 4.6 μM) was much lower than the MIC found for **2** (19.5 μg/mL = 48.6 μM) and other silver compounds containing TPMS, *viz*. [AgP_4_]·4TPMS·BF_4_ (16–32 μM) and [Ag(TPMS)P]·BF_4_ (128 μM) [30]. Concerning the other bacterial pathogens, compound **1** was still more active than **2** as shown, respectively, by the following values (in μM): *S. aureus* Newman, 34.5 and 128.1; *E. coli*, 17.8 and 36.1; and *B. contaminans*, 12.7 and 48.9. The activity of compound **2** towards the fungal strains *C. albicans* SC5134 and *C. glabrata* CBS138 (6.3 and 5.7 μg/mL = 15.7 μM and 14.2, respectively) is intermediate between those previously found for [AgP_4_]·4TPMS·BF_4_ (8 μM) and [Ag(TPMS)P]·BF_4_ (32 μM) [30]. Considering that integrity is retained in solution for the mononuclear [Ag(TPM^OH^)(NO_3_)] (obtained from **1**) and [Ag(TPMS)] (obtained from **2**), both neutral entities, the observed overall distinct biological behavior is conceivably related to steric effects, apart from the electronic ones. Under the assumption that the molecules of compound **1** are larger than those of **2**, based on the molecular masses of the compounds, the observed differences in MIC values towards Gram-negative bacteria cannot be explained based on the relative size of the compounds since the former should have greater difficulties in the crossing of the cell walls. Without doubt, compound **2** is less effective on bacteria than on fungi. 

The toxicity of compounds **1** and **2** was assessed using the nematode *C. elegans*, widely used to assess the toxicity of various compounds, as recently reviewed by Li et al. [40]. For this purpose, larvae of the *C. elegans* strain BN2 at the larval stage L4 were placed in M9 buffer containing heat-killed *E. coli* and supplemented with concentrations of compounds **1** or **2** up to 100 µg/mL, and the number of surviving worms was then registered after 24, 48, and 72 h (Figure 6). Results shown in Figure 6 show that after 24 h, 95 and 90% of the worms remained alive in the presence of 100 µM of compounds **1** or **2**, respectively. After 74 h of incubation in the presence of 100 µM of compounds **1** or **2**, about 40 and 70% of the worms were alive, respectively. Results from these experiments were also used to determine the concentrations of compounds **1** and **2** that killed 50% of the nematodes, LD_50_, at 48h and 72 h. Only for compound **1** at 72 h, the LD_50_ could be estimated as 73.2 µg/L, otherwise the LD_50_ was > 100 µg/L. Although further toxicity tests need to be performed in more complex animal models, these results are promising in showing a therapeutic window that can be exploited to use compounds **1** and **2** as antimicrobials.

Hirshfeld analyses [41] were performed on **1** and **2**, as well as on [AgP_4_].4TPMS.BF_4_, in order to explore intermolecular interactions with the aim of recognizing the structural features that may be related to the MIC activity of the compounds, without excluding the fact that TPMS in the silver phosphine complex do not act as ligands but as counter-anions instead. Crystal Explorer version 3.1 was used for the calculations [42]. The 2D fingerprint plots relating the distance from the surfaces to the nearest atom exterior (de) and the interior (di) to the surface are displayed in Appendix A. They enabled us (Figure 7, left) to identify the type of non-covalent interactions that mostly contribute to the surfaces, including H···H (33.4–53.4%), followed by O···H contact, at least for compounds **1** and **2** (over 30%, against only 5.6% for [AgP_4_].4TPMS.BF_4_. Contributions of Ag···N contacts are present only in compounds **1** (5.2%) and **2** (6.5%). Based on these general features, no possible explanation can be envisioned for the biological behavior of the compounds. However, believing that the biological behavior of the compounds is grounded in non-covalent interactions, the distances from the surfaces to the nearest atom exterior to the surface (de) may play a role. If so, the longer such distances, the higher the availability of the compound and its activity. Actually, the trend in O···H distances (Figure 7, right) follows the one observed for the activity against *S. aureus*, *E. coli*, and *P. aeruginosa*, where **1** > [AgP_4_].4TPMS.BF_4_ > **2** (see above). For *C. albicans*, however, it seems that the N···H contacts are the central ones. Despite the limited number of compounds of this study, such relationships are tempting, and they will conceivably inspire and guide future analyses on these matters.

Although structural features influence the antimicrobial effect of compounds **1** and **2**, their specific targets and molecular mechanisms of action remain to be studied. Future work on the unveiling of the mechanisms of action should be performed, including the evaluation of the effects of the compounds on oxidative stress including lipid peroxidation, analysis of membrane integrity, and electron transfer chain functioning, and thiol group oxidation, general mechanisms already described as underlying silver antimicrobial activity.

## 3. Materials and Methods

### 3.1. General Procedures

Reagents and solvents were obtained from commercial suppliers (AgNO_3_ ≥ 99%, Aldrich; methanol ≥ 99.8%, Fisher Chemical; DMSO, Fisher Chemical, Hampton, NH, USA) and used without further purification. All reactions were carried out in air. Compounds Li(TPMS), TPM^OH^, and [Ag(TPMS)] (**2**) were obtained and characterized following the reported procedures [37,43,44]. Infrared spectra (4000–400 cm^−1^) were recorded using the Cary 630 FTIR Spectrometer (Agilent Technologies, Santa Clara, CA, USA). The elemental analyses (C, N and H) were carried out by the Microanalytical Service (at Laboratório de Análises) at Instituto Superior Técnico (IST). NMR analysis was performed using the Bruker Advance (Bruker, Billerica, MA, USA) 300 MHz spectrometer at ambient temperature (23 °C). The chemical shifts were internally referenced to residual protio-solvent resonance and reported in ppm relative to tetramethyl silane.

### 3.2. Synthesis of [Ag(NO_3_)(μ-1κN;2κN′,N″-TPM^OH^)]_n_ (**1**)

In a round bottom flask, a solution of TPM^OH^ (100 mg, 0.4 mmol) in 10 mL of methanol was added to a solution of AgNO_3_ (68 mg, 0.4 mmol) in 10 mL of methanol. The obtained colorless solution underwent overnight stirring at room temperature, followed by evaporation in open air to obtain compound **1** as colorless crystals that were suitable for SCXRD analysis.

Yield = 58%, based on silver. The elemental analysis calculated (%) for C_11_H_12_AgN_7_O_4_·H_2_O (432.14 g/mol) was as follows: C 30.57, H 3.27, N 22.69; found: C 30.32, H 3.19, N 22.43. FTIR (KBr): ν (cm^−1^) = 3153 w, 1521 m, 1395 s, 1338 m, 1294 s, 1205 m, 1093 s, 1062 m, 1031 m, 983 w, 954 m, 916 m, 872 s, 822 w, 773 s, 748 s, 662 m, 620 s, 604 s, 505 m. ^1^H NMR (300 MHz, DMSO-*d*_6_, *δ*): 7.67 (d, *J* = 1.2 Hz, 3H, 5-*H*-pz), 7.52 (d, *J* = 2.4 Hz, 3H, 3-*H*-pz), 6.41 (dd, *J*_1_ = 2.3 Hz, *J*_2_ = 2.1 Hz, 3H, 4-*H*-pz), 6.00 (t, *J* = 5.7 Hz, 1H, O*H*), 4.96 (d, *J* = 6.0 Hz, 2H, C*H*_2_). ^13^C{^1^H} NMR (300 MHz, DMSO-*d*_6_, *δ*): 141.13 (3-*C*-pz), 131.15 (5-*C*-pz), 106.47 (4-*C*-pz), 89.85 (*C*CH_2_), 65.38 (*C*H_2_).

### 3.3. X-ray Structure Determination

The crystals of compound **1** were immersed in cryo-oil, mounted in a nylon loop. X-ray diffraction intensity data were then collected by a Bruker SMART APEX-II CCD area detector (Bruker, Billerica, MA, USA), using graphite monochromated MoKα radiation (λ = 0.71073 Å) at 296 K. SADABS was used for absorption corrections [45,46]. The structure was elucidated through direct methods and refined on *F*^2^ by the full-matrix least-squares method, employing Bruker’s SHELXTL-97 (Bruker, Billerica, MA, USA) [47]. Anisotropic refinement was performed for all non-hydrogen atoms. Comprehensive crystallographic data are summarized in Appendix A. The pertinent structural information has been deposited at the Cambridge Crystallographic Data Centre [CCDC 2332207]. Interested parties can obtain a complimentary copy of these data from the Director, CCDC, 12 Union Road, Cambridge CB2 1EZ, UK (Fax: (+44) 1223-336033; E-mail: deposit@ccdc.cam.ac.uk or www.ccdc.cam.ac.uk/data_request/cif, accessed on 1 July 2024.

### 3.4. Bacterial and Fungal Strains and Determination of Antimicrobial Activity

The bacterial strains, *S. aureus* Newman, *E. coli* ATCC25922, *B. contaminans* IST408 and *P. aeruginosa* 477, as well as the fungal strains, *C. glabrata* CBS138 and *C. albicans* SC5134, were used in this work. Bacterial and fungal strains were maintained, respectively, in Lennox Broth (LB) solid medium (10 g/L tryptone, 5 g/L yeast extract, 5 g/L NaCl and 15 g/L agar) and Yeast Extract–Peptone–Dextrose (YPD) solid medium (20 g/L glucose, 20 g/L peptone, 10 g/L yeast extract and 15 g/L agar).

Stock solutions of **1**, **2**, and TPM^OH^ (10.0, 5.0, and 10.0 mg/mL, respectively) were prepared in 100% DMSO.

Measurement of the antibacterial activity of the compounds was carried out with the standardized microdilution method recommended by EUCAST (European Committee on Antimicrobial Susceptibility Testing) [48]. Briefly, 96-well polystyrene microtiter plates (Greiner Bio-One, Kremsmünster, Austria) were filled with 100 μL of Mueller–Hinton (MH) broth (Fluka Analytical, Darmstadt, Germany). Sequential 1:2 dilutions of each compound from the stock solutions were performed to obtain final concentrations ranging from 250 µg/mL to 0.49 µg/mL. Bacterial cultures that had been grown for five hours (carried out in MHB at 37 °C and 250 rev·min^−1^) were diluted with MHB fresh medium and added to the 96-well plates at a final inoculum of 5 × 10^5^ CFU/mL. The microplates were then incubated for 22 h at 37 °C. After incubation, the wells were examined for turbidity (growth), resuspended by pipetting, and their optical density was measured in a SPECTROstar Nano microplate reader (BMG Labtech, Ortenberg, Germany) at 640 nm.

Antifungal susceptibility testing was carried out according to the standardized microdilution method recommended by EUCAST (European Committee on Antimicrobial Susceptibility Testing) for *Candida* spp. [49]. Briefly, the 96-well microtiter plates (Greiner Bio-One) were filled with 100 μL of RPMI-1640 liquid medium (SIGMA, Burlington, MA, USA) buffered to pH 7.0 with 0.165 M morpholine propanesulfonic acid (MOPS; SIGMA). Sequential 1:2 dilutions of each compound stock solution were carried out in order to obtain final concentrations ranging from 125 µg/mL to 0.49 µg/mL. Overnight grown fungal cultures (carried out in YPD broth at 30 °C and 250 rev.min^−1^) were diluted with RPMI-1640 fresh liquid medium to a final optical density of 0.025, measured at 530 nm (OD_530_) in a Hitachi U-2000 UV/Vis spectrophotometer (Hitachi, Tokyo, Japan). The wells were then inoculated with the addition of 100 μL of the fungal suspensions (*C. glabrata* CBS138 or *C. albicans* SC5134) and incubated for 24 h at 35 °C. After incubation, the wells were examined for turbidity (growth), resuspended, and their optical density was measured in a SPECTROstar Nano microplate reader (BMG Labtech) at 530 nm. 

Adequate quality controls were performed using ampicillin for the bacterial strains and fluconazole and auranofin for the fungal strains, respectively.

All compounds were tested in at least three independent experiments and in duplicate wells. Minimum inhibitory concentration (MIC) was estimated after data fitting of the OD_640_ or OD_530_ mean values using a modified Gompertz equation, using the GraphPad Prism software (version 6.07) [50]. In each experiment, positive (without compound) and negative controls (no organism inoculum) were carried out. The effect of 5% (*v*/*v*) or 2.5% (*v*/*v*) DMSO on bacterial or fungal growth, respectively, was also assessed, and no effects were detected.

### 3.5. Toxicity of Complexes to the Nematode Caenorhabditis Elegans 

The nematode *C. elegans* Bristol strain N2 used in the present work was obtained from the Caenorhabditis Genetics Center (Minneapolis, MN, USA), which is supported by the National Institutes of Health, Office of Research Infrastructure Programs (P40 OD010440).

The nematode strain was maintained at 20 °C on Nematode Growth Medium (NGM) agar plates seeded with *E. coli* OP50 and synchronized using standard protocols and as previously described [51]. The NGM was composed of 25 mM NaCl, 1.7% (*w*/*v*) agar, 2.5 mg/mL peptone, 5 μg/mL cholesterol, 1 mM CaCl_2_, 1 mM MgSO_4_, and 50 mM KH_2_PO_4_ (pH 6.0). The uracil-deficient *E. coli* OP50 was used as the nematode food source. L4 synchronized worms were exposed to complexes final concentrations of 0, 10, 25, 50, 75, or 100 µg/mL in M9 buffer (pH 6), supplemented with 0.2% (*w*/*v*) of heat-killed *E. coli* OP50. Nematode appearance and viability were followed for 72 h at 20 °C, in a final volume of 500 µL contained in the wells of a 24-well plate. The total number of living and dead worms per well was assessed at 24, 48, and 72 h timepoints with the aid of a Zeiss Stemi 2000-C stereomicroscope (Zeiss, Jena, Germany). The survival of live worms was computed by tapping the plate and counting the moving worms. At least three independent experiments were performed using a total of 198 ± 28 L4 worms per each experimental condition tested. Kaplan–Meier survival curves were drawn using GraphPad Prism (GraphPad Software, San Diego, CA, USA). Concentrations that resulted in the death of 50% of the tested organisms (lethal concentration—LC_50_) were determined using the LC_50_ calculator (https://www.aatbio.com/tools/lc50-calculator, accessed on 1 July 2024), using a four-parameter logistic model with background correction and normalization.

## 4. Conclusions

The novel hydrosoluble silver complex [Ag(NO_3_)(μ-1κ*N*;2*κN*′,*N*″-TPM^OH^)]_n_ (**1**) was synthesized and characterized. SCXRD analysis proved compound **1** as a coordination polymer with bridging TPM^OH^ ligands in the solid state. The solubility of compound **1** in polar solvents and the observed NMR spectra indicate a monomeric complex in solution. The antimicrobial activities of compound **1** and of the [Ag(μ-1κ*N*;2*κN*′,*N*″-TPMS)]_n_ (**2**) analogue were evaluated against a set of Gram-negative and Gram-positive bacterial pathogens, and the two important fungal pathogens, *C. albicans* and *C. glabrata*. Compound **1** showed remarkably low MIC values towards the Gram-negative strains tested (particularly in the case of *P. aeruginosa* 477 (4.6 μM)), in comparison to **2** and the other TPMS-containing silver species; however, it had a lower activity towards the Gram-positive *S. aureus* and the fungal species *C. albicans* and *C. glabrata*. Remarkably high LC_50_ values were obtained using living *C. elegans*, highlighting the potential therapeutic utility of compounds **1** and **2**. Previous work from our research group has highlighted silver camphorimine complexes as a group of novel compounds with antimicrobial activities against pathogenic bacteria and fungi [52,53,54]. The present work adds silver C-scorpionate complexes to the silver complexes with antimicrobial activities of potential therapeutic use in medicine. 

## Figures and Tables

**Figure 1 antibiotics-13-00647-f001:**
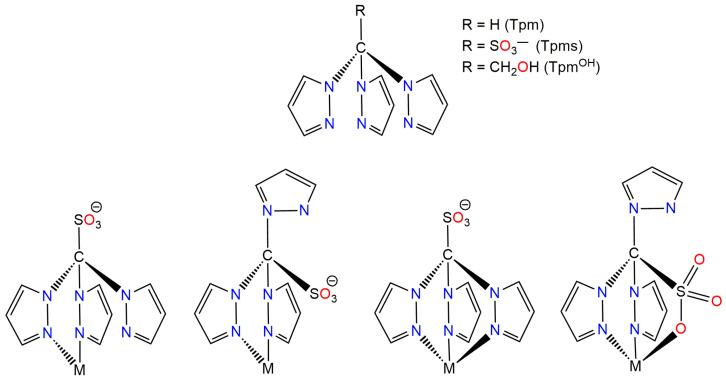
(**Top**): schematic representation of TPM, TPMS, and TPM^OH^; (**Bottom**): bipodal and tripodal coordination ability of TPMS.

**Figure 2 antibiotics-13-00647-f002:**
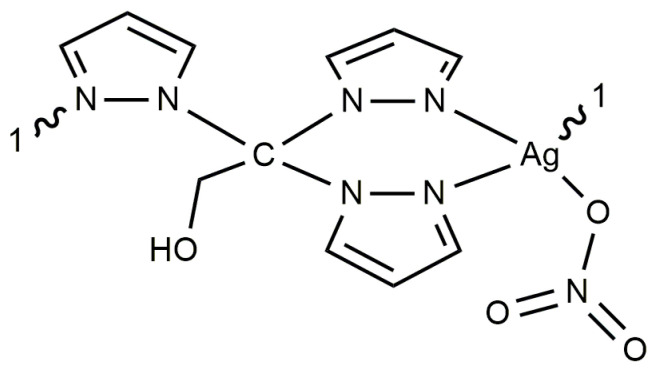
Polymeric structure of compound **1** in solid state.

**Figure 3 antibiotics-13-00647-f003:**
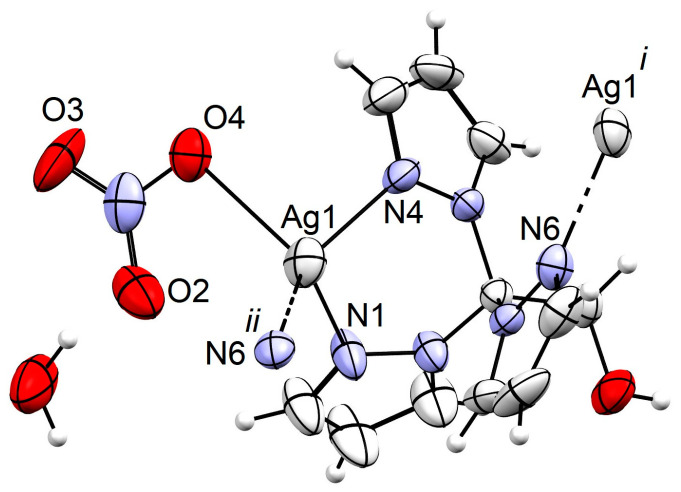
ORTEP diagram of compound **1**, drawn at a 40% probability level, with partial atom labelling scheme. Selected bond distances (Å) and angles (°): Ag1–N1 2.452(4), Ag1–N4 2.264(6), Ag1–N6*^ii^* 2.231(6), Ag1–O4A 2.57(4), Ag1···Ag1*^i^* 5.0830(6), N1–Ag1–N6*^ii^* 108.5(2), N4–Ag1–N6*^ii^* 149.3(2), N1–Ag1–N4 77.4(2), O4A–Ag1–N6*^ii^* 121.5(9), O4A–Ag1–N1 103.9(7) and O4A–Ag1–N4 84.3(7). Symmetry operations to generate the equivalent atoms: (*i*) −1/2 + x,1/2 − y, −1/2 + z; (*ii*) 1/2 + x,1/2 − y,1/2 + z.

**Figure 4 antibiotics-13-00647-f004:**
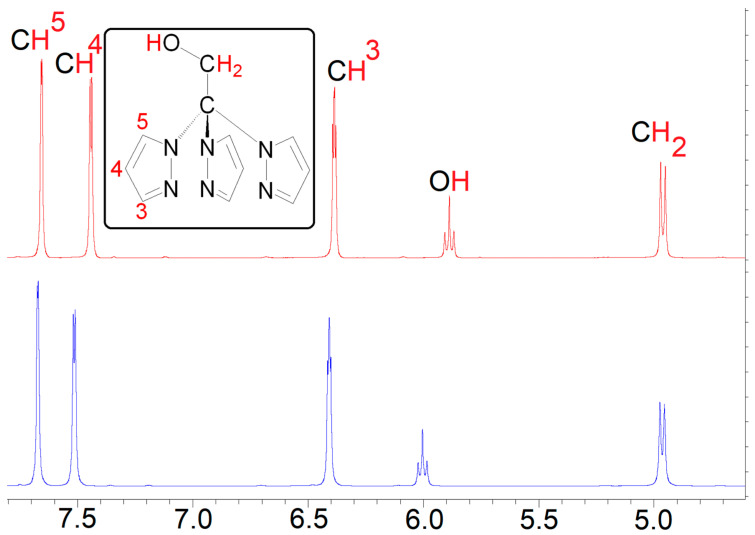
^1^H NMR spectrum of TPM^OH^ (**top**) and complex **1** (**bottom**) in DMSO-*d*_6_.

**Figure 5 antibiotics-13-00647-f005:**
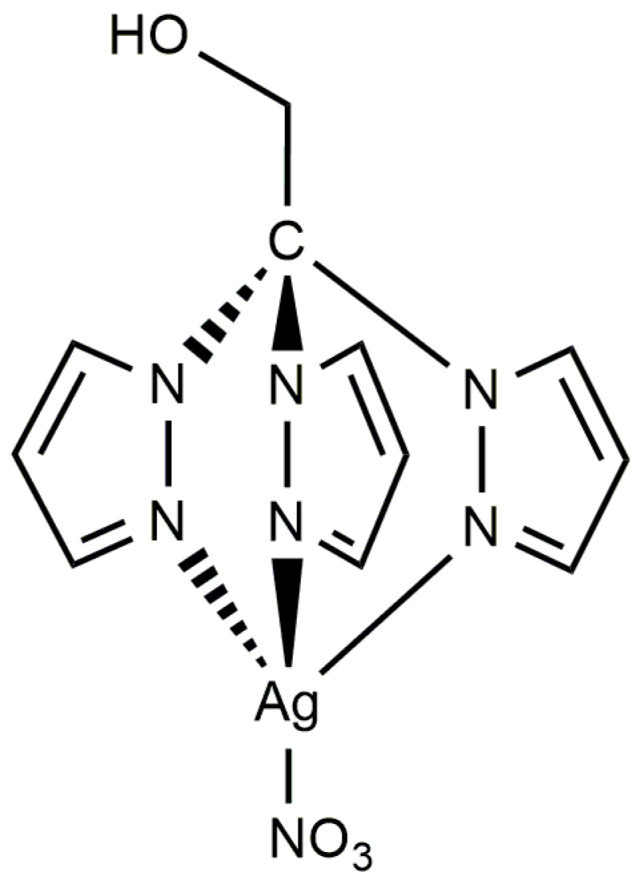
Structure of complex **1** in solution.

**Figure 6 antibiotics-13-00647-f006:**
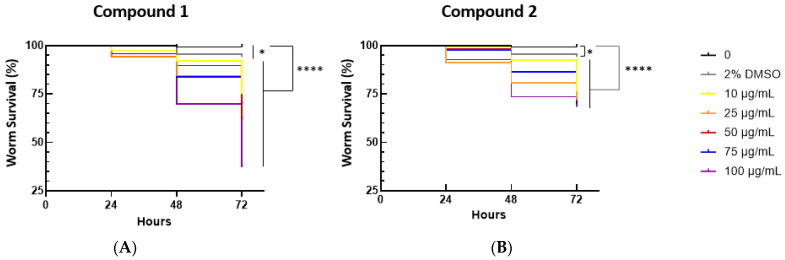
Percentage of surviving worms after incubation in M9 buffer containing 0.2% (*w*/*v*) of heat-killed *E. coli* OP50 as food source and supplemented with the indicated concentrations of compounds **1** (**A**) or **2** (**B**). Controls without compounds **1** and **2** or with DMSO were used as controls. Results are the means of at least three independent experiments using a total of 198 ± 28 worms at the L4 stage of development per each experimental condition tested. Differences compared to the control were considered as significant, with an * indicating a *p* < 0.0165, and **** indicating a *p* < 0.000001, determined using the log-rank (Mantel–Cox) test.

**Figure 7 antibiotics-13-00647-f007:**
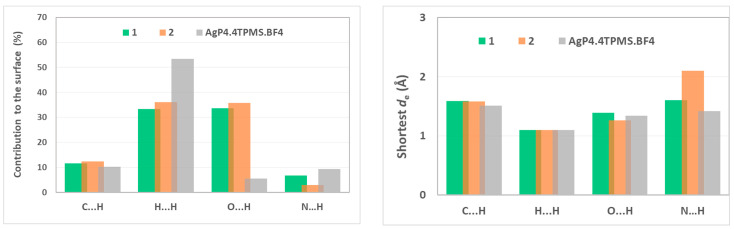
(**Left**): Percentage contribution to the Hirshfeld surface of compounds **1**, **2,** and [AgP_4_].4TPMS.BF_4_. (**Right**): Shortest *d*_e_ dimensions (in Å) in the most relevant non-covalent interactions detected in compounds **1**, **2** and [AgP_4_].4TPMS.BF_4_.

**Table 1 antibiotics-13-00647-t001:** Minimal inhibitory concentration (MIC) for compounds **1**, **2,** and TPM^OH^ towards the indicated bacterial and fungal strains. Results are presented as the means ± SD of at least three experiments carried out in duplicate.

		MIC (μg/mL)	
	1	2	TPM^OH^
*E. coli* ATCC25922	7.7 ± 0.2	14.5 ± 4.3	>250
*P. aeruginosa* 477	2.0 ± 0.1	19.5 ± 0.9	>250
*B. contaminans* IST408	5.5 ± 1.3	19.6 ± 4.0	>250
*S. aureus* Newman	17.9 ± 3.2	51.4 ± 8.7	>250
*C. albicans* SC5134	15.4 ± 0.4	6.3 ± 1.0	>125
*C. glabrata* CBS138	31.5 ± 0.1	5.7 ± 0.2	>125

## Data Availability

The data presented in this study are available in article and Appendix A.

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
