# Peer review of "Antimicrobial Activity of Water-Soluble Silver Complexes Bearing C-Scorpionate Ligands"

_antibiotics, 2024, doi:10.3390/antibiotics13070647_

Round 1
Reviewer 1 Report
Comments and Suggestions for Authors
The presents study’’ Antimicrobial activity of water-soluble silver complexes bear- 2 ing C-scorpionate ligands’’ report the synthesis and characterization of the novel silver coordination polymer bearing the TPMOH ligand, [Ag(NO3)(m-1kN;2kNʹ,Nʹʹ-TPMOH)]n (1). The antimicrobial activity of the silver-TPMS and -TPMOH complexes was investigated against Escherichia coli, Pseudomonas aeruginosa, Burkholderia ,Staphylococcus aureus, and the fungal species Candida albicans and C. glabrata. The finding indicated that, the compound has antimicrobial properties toward tested bacteria. While the topic is interesting and the effort is commendable. The study lack of critical and vital information, without them is not suitable for publication
Comment to the authors
1. The study was done without toxicity investigation which mean the safety concern is rise. Therefore, the toxicity test is required.
2. There is massive problem with MIC test. The CLSI document cited is largely out-dated.
There is no mentioning of QC strains used which leads to the assumption that the testing was done without quality controls
3. Line 158: ‘’ Reagents and solvents were obtained from commercial sources and used without further purification’’. More characterization test is required to confirm the compound purity and it chemical characterization.
4. The manuscript has high level of textual overlap with previous literature with 42% which means the plagiarism concerns is rise
Author Response
Reviewer #1
The presents study’’ Antimicrobial activity of water-soluble silver complexes bearing C-scorpionate ligands’’ report the synthesis and characterization of the novel silver coordination polymer bearing the TPMOH ligand, [Ag(NO3)(m-1kN;2kNʹ,Nʹʹ-TPMOH)]n (1). The antimicrobial activity of the silver-TPMS and -TPMOH complexes was investigated against Escherichia coli, Pseudomonas aeruginosa, Burkholderia contaminans, Staphylococcus aureus, and the fungal species Candida albicans and C. glabrata. The finding indicated that, the compound has antimicrobial properties toward tested bacteria. While the topic is interesting and the effort is commendable. The study lack of critical and vital information, without them is not suitable for publication.
Question: The study was done without toxicity investigation which mean the safety concern is rise. Therefore, the toxicity test is required.
Answer: The authors appreciate the observation and agree with the reviewer. We have performed toxicity tests of the compounds. For this purpose, we used the Caenorhabditis elegans as a toxicity animal model and followed the survival of worms at the larval stage L4 for 72 in the presence of concentrations of compounds up to 100 micromolar. Kaplan-Meyer survival plots are shown, as well as the estimated LD50 values for 24, 48 and 72 h, for complexes 1 and 2. Therefore, the results were added to the Abstract (new lines 29-32), Introduction (new lines 105-107), Results and discussion (new lines 173-200), Methods (new lines 317-339) and Conclusions (new lines 352-353).
Question: there is massive problem with MIC test. The CLSI document cited is largely out-dated.
There is no mentioning of QC strains used which leads to the assumption that the testing was done without quality controls
Answer: Thanks for the comment. In fact, as quality control we determined the MICs of ampicillin for the bacterial strains, and of fluconazole for the Candida strains. While the E. coli strain is a reference strain, the other bacteria are not reference strains. Therefore, we determined the MIC values for all bacterial strains and confirmed that the values were identical to previous values determined by us for the strains under study. Similarly, we used fluconazole and auranofin as reference for the fungal strains and confirmed that the MIC values were identical to values obtained in previous studies. We haven´t mentioned these experiments since the compounds under study are quite different from conventional antibiotics and no reference values are available. In order to meet the criticism, we have added in the Materials and methods section the following sentence in new lines 307-308: “Adequate quality controls were performed using ampicillin for the bacterial strains and fluconazole and auranofin for the fungal strains, respectively”.
Question: Line 158: ‘’ Reagents and solvents were obtained from commercial sources and used without further purification’’. More characterization test is required to confirm the compound purity and it chemical characterization.
Answer: Thanks for the observation. The commercial sources of the chemicals (reagents and solvents) used in the study were added. All technical data, including their purity, can be obtained from the material datasheet published on the corresponding websites. Considering the chemicals and known compounds used as precursors for this study, they were obtained and characterized following the published procedures, as stated at lines 235-237.
Question: The manuscript has high level of textual overlap with previous literature with 42% which means the plagiarism concerns is rise-
Answer: Thanks for the observation. Most of the problems arise because the methodologies used are standardized and routinely used in the laboratories and are widely used. Nevertheless, we have made a considerable effort to rewrite methodologies.
Reviewer 2 Report
Comments and Suggestions for Authors
The manuscript is titled "Antimicrobial activity of water-soluble silver complexes bearing C-scorpionate ligands ". A silver coordination polymer compound that is soluble in water was synthesized and characterized. Representative strains of the Gram-negative bacteria Escherichia coli, Pseudomonas aeruginosa, Burkholderia contaminans, the Gram-positive Staphylococcus aureus, Candida albicans, and C. glabrata were utilized to assess the antimicrobial activity of the silver-TPMS and -TPMOH complexes. However, it is crucial to offer further clarification on certain issues Therefore, it is recommended that this manuscript be published with only minor revisions.
The complete name of DMSO should be specified in line 93.
SCXRD should be specified with its comprehensive name in line 95, after which it may be referred to as SCXRD.
Despite being labeled "Results and Discussion" in the article's title, the discussion section appears to be quite inadequate. It is advised that you expand on this section.
Comments on the Quality of English Language
Minor revisions to the English language are necessary.
Author Response
Reviewer #2
The manuscript is titled "Antimicrobial activity of water-soluble silver complexes bearing C-scorpionate ligands ". A silver coordination polymer compound that is soluble in water was synthesized and characterized. Representative strains of the Gram-negative bacteria Escherichia coli, Pseudomonas aeruginosa, Burkholderia contaminans, the Gram-positive Staphylococcus aureus, Candida albicans, and C. glabrata were utilized to assess the antimicrobial activity of the silver-TPMS and -TPMOH complexes. However, it is crucial to offer further clarification on certain issues Therefore, it is recommended that this manuscript be published with only minor revisions.
Question: The complete name of DMSO should be specified in line 93.
Answer: Thanks for the observation. The complete name of DMSO was inserted the first time it appears in the text.
Question: SCXRD should be specified with its comprehensive name in line 95, after which it may be referred to as SCXRD.
Answer: Thanks for the observation. The complete name of SCXRD was inserted the first time it appears in the text.
Question: Despite being labeled "Results and Discussion" in the article's title, the discussion section appears to be quite inadequate. It is advised that you expand on this section.
Answer: Thanks for the observation. The section was reviewed and expanded as requested. A discussion of structure-activity relationships through Hirshfeld surfaces study was included, the relationship between the MIC values and the LD50 values found were discussed, as well as future directions of the work.
Reviewer 3 Report
Comments and Suggestions for Authors
In the Introduction, the author should explain the significance of the silver and silver complexes advantages.
When compared to other complex compounds, the reason why silver-TPMS is active in microbial activity. Could you explain its unique mechanism structure-property?
To improve the manuscript in a significant manner, it would be better to draw mechanistic figures for the manuscript.
Explain the role of lithium salt TPMS added to along with AgNO3
For table S1: Provide the XRD in figure format
The author should check the spelling and grammatical mistakes throughout the manuscript
Line no 134: The author must provide a comprehensive explanation of the bacteria, focusing on the strong antibacterial activity they possess compared to other bacterial strains.
Line no 166: the author should mention the specific temperature
Mention the slow evaporation method used in this SCXRD analysis
Line no182: Explain in detail about the X-ray structure determination
Author Response
Reviewer #3
Question: In the Introduction, the author should explain the significance of the silver and silver complexes advantages.
Answer: We thank the reviewer for the suggestion. A new paragraph was added into the introduction section (see new lines 50-56).
Question: When compared to other complex compounds, the reason why silver-TPMS is active in microbial activity. Could you explain its unique mechanism structure-property?
Answer: Thank you for the question. The mechanisms of action of the compounds was not addressed in this work. The structure-activity relationship was discussed based on Hirshfeld analysis. However, this analysis is limited and definitive conclusions need to consider also specificities from the microorganisms studies are also determinant for the activity of the compounds. We advance that some mechanisms worth to investigate in future studies might involve the action of reactive oxygen species and interference with membrane integrity and functions, as previously shown for silver complexes and silver nanoparticles.
Question: To improve the manuscript in a significant manner, it would be better to draw mechanistic figures for the manuscript.
Answer: Thanks for the suggestion. As the mechanisms underlying the biological activity of the compounds were not addressed in the present work, we decided not to speculate on this subject.
Question: Explain the role of lithium salt TPMS added to along with AgNO3
Answer: Thanks for the question. Lithium salt of Tpms is just a reactant that contains the Tpms ligand, which reacted with AgNO3 to give complex [Ag(TPMS)]n (2) following the published procedure, as explained in lines 103-105. It doesn’t have any other roles in our study.
Question: For table S1: Provide the XRD in figure format.
Answer: Thanks for the suggestion. Table S1 is showing some crystallographic data and structure refinement details for compound 1. The figure of the corresponding crystal structure of the compound, obtained through X-ray diffraction analysis, is given in the manuscript (Figure 3). We are not sure if this is what the Referee is asking for. If not, then we would appreciate if we can get some more details to answer this requirement properly.
Question: The author should check the spelling and grammatical mistakes throughout the manuscript.
Answer: Thanks for the suggestion. We have thoroughly checked grammatical mistakes and typos.
Question: Line no 134: The author must provide a comprehensive explanation of the bacteria, focusing on the strong antibacterial activity they possess compared to other bacterial strains.
Answer: Thanks for the suggestion, although we haven´t really understood the suggestion. The bacterial strains were selected for being representatives of problematic species as indicated in the Introduction. They belong to the priority list of WHO, as explained in the Introduction. Another aspect is that the selected strains are not particularly sensitive to antibiotics. For instance B. contaminans is resistant to several antibiotics.
Question: Line no 166: the author should mention the specific temperature
Answer: Thank you for the observation. The temperature was added.
Question: Mention the slow evaporation method used in this SCXRD analysis
Answer: The reaction mixture used to obtain the compound was simply left for evaporation in open air, as explained in lines 210 and 211, with no other steps.
Question: Line no182: Explain in detail about the X-ray structure determination.
Answer: Thanks for the suggestion. All details about SCXRD analysis are mentioned at section 3.3 (X-ray structure determination), including the method used to obtain the intensity data, type of diffractometer, detector, radiation, temperature, absorption corrections and structural refinements. The preparation of crystals in cryo-oil and its fixation on a Nylon loop for the measurements was added to that section.
Reviewer 4 Report
Comments and Suggestions for Authors
The manuscript entitled: “Antimicrobial activity of water-soluble silver complexes bear-2 ing C-scorpionate ligands”, is a interesting manuscript, as it describes and characterizes two relevant compounds.
Nevertheless, in my point of view, the discussion and conclusions of this manuscript fall short in terms of potential and impact. Major issues that I would like to underscore:
1. In my understanding the author perform little discussion of the possible modus operandi of the compounds according to the distinctive traits of each tested microorganisms.
2. In addition, and I am sorry if I missed it, but I did not clerarly observed any referenc to the feasibility of using these compounds considering their MIC.
3. Furthermore, the authors just slightly mention the cytotoxicity of similar compounds in the introduction.
4. Finally, I failed to observe any mentions to the importance of absence of cytotoxicity in the discussion section, and how the author intends to tackle this.
Minor comments:
Species names, the first appearance (in the abstract and in the text body) of species name it must include its complete name, genus and specific epithet. After its first appearance the guns should be consistently abbreviated. This should be performed for every microorganism, including, Candida glabrata (first appearance), C. glabrata for following appearances.
How were figure 1 to 3 obtained? Please specify the software and including other relevant information if required.
Figure 5, please specify which type of solutions.
Line 136, Candida, as genus should be italicized.
Line 150, “Gran negative”, please revise typo.
Author Response
Reviewer #4
The manuscript entitled: “Antimicrobial activity of water-soluble silver complexes bear-2 ing C-scorpionate ligands”, is a interesting manuscript, as it describes and characterizes two relevant compounds.
Nevertheless, in my point of view, the discussion and conclusions of this manuscript fall short in terms of potential and impact. Major issues that I would like to underscore:
Question: In my understanding the author perform little discussion of the possible modus operandi of the compounds according to the distinctive traits of each tested microorganisms.
Answer: Thanks for the comment. The mechanisms underlying the antimicrobial activity were not addressed in this work and therefore we haven´t speculate on this subject. Nevertheless, some future directions of the work focusing on mechanisms was introduced, namely formation of reactive oxygen species due to silver, and effects of silver on membrane integrity and function.
Question: In addition, and I am sorry if I missed it, but I did not clerarly observed any referenc to the feasibility of using these compounds considering their MIC.
Answer: Thanks for the observation. In fact were haven’t mentioned the feasibility of using these compounds as toxicity was not assessed. We have now performed toxicity tests in the revised version and the LD50 values found were quite higher than the MIC, with a wide range of concentrations that can be used. This is now mentioned in the revised version of the manuscript.
Question: Furthermore, the authors just slightly mention the cytotoxicity of similar compounds in the introduction.
Answer: Thanks for the observation. As already mentioned, we have performed toxicity tests using the nematode C. elegans, and results were added to the revised manuscript.
Question: Finally, I failed to observe any mentions to the importance of absence of cytotoxicity in the discussion section, and how the author intends to tackle this.
Answer: Thanks. As already mentioned, toxicity tests were performed and results are now included in the revised manuscript.
Question: Species names, the first appearance (in the abstract and in the text body) of species name it must include its complete name, genus and specific epithet. After its first appearance the guns should be consistently abbreviated. This should be performed for every microorganism, including, Candida glabrata (first appearance), C. glabrata for following appearances.
Answer: The manuscript was thoroughly revised and the names of species were adequately corrected.
Question: How were figure 1 to 3 obtained? Please specify the software and including other relevant information if required.
Answer: Thanks for the question. The molecular structures in Figures 1 and 2 were obtained using ChemDraw software, which is a common tool to draw chemical structures. Figure 3 was obtained by Mercury, which is a freeware developed by the Cambridge Crystallographic Data Centre, based on the “CIF” file of the compound.
Question: Figure 5, please specify which type of solutions.
Answer: Thanks for the observation. Figure 5 represents the structure of complex 1 in solutions of polar solvents, such as water and DMSO, as suggested in line 128 and proved at the same paragraph using the NMR study (lines 128-139).
Question: Line 136, Candida, as genus should be italicized.
Answer: Thanks for the observation. Corrected.
Question: Line 150, “Gran negative”, please revise typo.
Answer: Thanks for the observation. The typo was corrected.
Round 2
Reviewer 4 Report
Comments and Suggestions for Authors
The manuscript entitled: “Antimicrobial activity of water-soluble silver complexes bearing C-scorpionate ligands”
Line 176, please add the set of references.
Figure 6, I beg your pardon, but I do not understand the results displayed in this figure. Ate all samples as lines within the same column? May a I suggest a single column for each concentration and for each time point? Similar to Figure 7. In my opinion these results are not clear.
Line 178 the authors mention: “liquid medium” however in line 194 refer M9 buffer. Please clarify. I would like to mention that usually the survival is “much harder” in buffer in comparison to medium. If that is the case, in my opinion, this should be highlighted by the authors.
Figure 7, are the results displayed in the figures the same as in the table? If so, that corresponds to data duplication. If the authors wish to include the numerical values please inset them above each column.
Line 323, please include the reference line 323.
Is the conclusions section in the adequate order of the manuscript?
Author Response
Answers to reviewers:
Question: Line 176, please add the set of references.
Answer: We thank the reviewer for the suggestion. We have already used a review reference on the use of C. elegans as a model organism in toxicity testing. The sentences were: “The toxicity of compounds 1 and 2 was assessed using the nematode C. elegans. The worm has been widely used to assess the toxicity of various compounds [40].
In the revised version of the manuscript, we fused the sentences to evidence the revision used a reference. It now reads as follows:
The toxicity of compounds 1 and 2 was assessed using the nematode C. elegans, widely used to assess the toxicity of various compounds as recently reviewed by Li et al. [40].
Question: Figure 6, I beg your pardon, but I do not understand the results displayed in this figure. Ate all samples as lines within the same column? May a I suggest a single column for each concentration and for each time point? Similar to Figure 7. In my opinion these results are not clear.
Answer: Figure 6 is a Kaplan- Meyer representation of the percentage survival of the worms populations, a standard and widely used mode of representation of survival of individuals in a population. Maybe the reviewer is not familiar with this type of representation where each line represents a condition (in this case Compound 1 or 2 different concentrations, and the percentage of survival after 24, 48 or 72 h), and as stated, at least 198 worms were used per experiment. Statistics was applied by comparing each condition with the control. Side lines indicate the level of significance of results, as indicated by an * or ****. Furthermore, a bar representation would be too confusing.
Question: Line 178 the authors mention: “liquid medium” however in line 194 refer M9 buffer. Please clarify. I would like to mention that usually the survival is “much harder” in buffer in comparison to medium. If that is the case, in my opinion, this should be highlighted by the authors.
Answer: Thanks for the observation. In fact, we used M9 buffer. The modification was done and the new sentence now reads as follows: “For this purpose, larvae of the C. elegans strain BN2 at the larval stage L4 were placed in M9 buffer containing heat-killed E. coli and supplemented with concentrations of 1 or 2 up to 100 µg/mL, and the number of surviving worms was registered after 24, 48 and 72 h (Fig. 6).”.
Question: Figure 7, are the results displayed in the figures the same as in the table? If so, that corresponds to data duplication. If the authors wish to include the numerical values please inset them above each column.
Answer: Thanks for the observation. In fact, the results are the same, and we decided to delete the number in the Table and not to include them on the top of bars, otherwise the Figure would turn confusing.
Question: Line 323, please include the reference line 323.
Answer: Reference [51] is already included in the reference list.
Question: Is the conclusions section in the adequate order of the manuscript?
Answer: Thanks for the observation. We followed the order of the template available at the journal webpage, were Conclusions come after Materials and Methods.
Additional Corrections:
In addition to the reviewers comments, we have also made the following minor changes:
A space was added between the following words: line 59“ …[14]. This”; line 64 “…carbenes [18-20]; line 65: “…carboxylates [21-23].
Line 95: IC50 was formatted to “IC50”
Line 180: Fig. 6 was substituted by “Figure 6”.
Line 185: compound 1; the “1” was formatted in bold.
Line 217: Figure 6 was corrected to Figure 7.
Line 224: Figure 6 was corrected to Figure 7.
Line 237: “reaction” was corrected to “reactions”
Line 282: “Antibacterial susceptibility" was corrected to “Antibacterial activity”
Line 298: “compound solution” was substituted by “compound stock solution”.
Round 3
Reviewer 4 Report
Comments and Suggestions for Authors
I am satisfied with the performed improvments.